# Synergistic Activity of DNA Damage Response Inhibitors in Combination with Radium-223 in Prostate Cancer

**DOI:** 10.3390/cancers16081510

**Published:** 2024-04-15

**Authors:** Victoria L. Dunne, Timothy C. Wright, Francisco D. C. Guerra Liberal, Joe M. O’Sullivan, Kevin M. Prise

**Affiliations:** 1Patrick G Johnston Centre for Cancer Research, Queen’s University Belfast, Belfast BT9 7AE, UK; twright03@qub.ac.uk (T.C.W.); f.liberal@qub.ac.uk (F.D.C.G.L.); joe.osullivan@qub.ac.uk (J.M.O.); k.prise@qub.ac.uk (K.M.P.); 2Northern Ireland Cancer Centre, Belfast Health and Social Care Trust, Belfast BT9 7AB, UK

**Keywords:** prostate cancer, radium-223, X-rays, ATR inhibition, ATM inhibition, PARP inhibition, DNA damage, cell cycle checkpoints

## Abstract

**Simple Summary:**

Radium-223 is used for the management of metastatic castration-resistant prostate cancer. However, the response to this treatment is poor. Targeting the DNA damage response pathway through pharmacological inhibition has the ability to enhance the radiation response. In this study, we investigated the radiosensitising effects of three different DNA damage response inhibitors targeting ATM, ATR, and PARP in combination with X-rays and radium-223 on human prostate cancer cell lines. Their effects on cell survival, DNA double-strand break repair, cell cycle distribution, and apoptosis were evaluated. We determined that these inhibitors increase the ability of X-rays and radium-223 to kill prostate cells to varying degrees. Moreover, inhibition of the DNA damage response pathways impeded the repair of radiation-induced DNA double-strand breaks and induced changes in cell cycle distribution. Altogether, our study determined DDR inhibition as a promising strategy to increase the effectiveness of treatments utilising X-rays and/or ^223^Ra for localised and metastatic castration-resistant prostate cancer.

**Abstract:**

Radium-223 (^223^Ra) and Lutetium-177-labelled-PSMA-617 (^177^Lu-PSMA) are currently the only radiopharmaceutical treatments to prolong survival for patients with metastatic-castration-resistant prostate cancer (mCRPC); however, mCRPC remains an aggressive disease. Recent clinical evidence suggests patients with mutations in DNA repair genes associated with homologous recombination have a greater clinical benefit from ^223^Ra. In this study, we aimed to determine the utility of combining DNA damage response (DDR) inhibitors to increase the therapeutic efficacy of X-rays, or ^223^Ra. Radiobiological responses were characterised by in vitro assessment of clonogenic survival, repair of double strand breaks, cell cycle distribution, and apoptosis via PARP-1 cleavage. Here, we show that DDR inhibitors increase the therapeutic efficacy of both radiation qualities examined, which is associated with greater levels of residual DNA damage. Co-treatment of ATM or PARP inhibition with ^223^Ra increased cell cycle arrest in the G2/M phase. In comparison, combined ATR inhibition and radiation qualities caused G2/M checkpoint abrogation. Additionally, greater levels of apoptosis were observed after the combination of DDR inhibitors with ^223^Ra. This study identified the ATR inhibitor as the most synergistic inhibitor for both radiation qualities, supporting further pre-clinical evaluation of DDR inhibitors in combination with ^223^Ra for the treatment of prostate cancer.

## 1. Introduction

Prostate cancer is the most common cancer in men and the fifth leading cause of cancer deaths in 2020 worldwide [1]. External beam radiotherapy (RT) is one of the mainstay therapeutic strategies for the management of localised and locally advanced prostate cancer. Despite recent prostate cancer treatment advances, radioresistance contributes to recurrence in 30–40% of patients, and ultimately, a large subset of these patients will be diagnosed with metastatic-castration-resistant prostate cancer (mCRPC). Additionally, the incidence of men presenting with mCRPC at the time of first diagnosis from 2008 to 2018 has increased by 5% [2,3,4].

In 2013, the FDA approved radium-223 (^223^Ra), a targeted α-particle-emitting radionuclide, to treat patients with mCRPC, based on results from the phase III Alpharadin in Symptomatic Prostate Cancer Patients (ALSYMPCA) trial [5]. ^223^Ra is a calcium-mimetic isotope that selectively binds to mineral hydroxyapatite in areas of increased metabolic activity, such as bone metastases [6]. The anti-tumour efficacy of ^223^Ra to prolong patient survival is due to its high linear energy transfer (LET) α-particle radiation, which results in multiple and complex double-strand DNA breaks (DSBs), which are increasingly resistant to cellular repair mechanisms [7,8]. In comparison, low-LET X-rays are less effective at killing cells as a result of the sparse pattern of ionisations deposited; therefore, DSBs are more efficiently repaired. However, despite the clinical success of ^223^RaCl_2_, mCRPC remains incurable, with a 5-year survival rate of 30% [9].

To maintain genomic stability, cancer cells respond to radiation-induced DNA damage by activating a dynamic network of DNA damage response (DDR) signalling pathways and cell-cycle checkpoints. Ataxia-telangiectasia mutated (ATM) and Rad3-related (ATR) are members of the phosphatidyl inositol 3-kinase-like kinases (PIKKs) family and are fundamental in the detection, signalling, and repair of DNA damage [7]. The specific kinase activated is dependent on the type of DNA lesion, as ATM responds to DSBs and ATR responds to single-stranded DNA caused by replication stress [7,10]. These kinases are also involved in triggering DNA damage checkpoints, apoptosis, and senescence in response to radiation [11]. Furthermore, Mladenova et al. (2021) have recently demonstrated an enhanced ATR dependence for the activation of the G2 checkpoint after exposure to high LET radiation [12]. Preclinical studies have demonstrated a synthetically lethal interaction between ATM and ATR, whereby ATR inhibition selectively leads to cell death in cells with defective ATM activity [13].

The nuclear proteins poly (ADP-ribose) polymerase-1 (PARP-1) and polymerase-2 (PARP-2) are also associated with a number of DDR-related processes. Specifically, PARP-1 mediates single-strand DNA break repair [8], alternative end-joining of DSBs [14], and also plays a key role in homologous recombination repair (HRR) [15]. Consequently, pharmacologic inhibition of PARP in cells with genetic or functional defects in HRR genes, including BRCA1/2, has proven to be synthetically lethal [16,17]. Recently, the PARP inhibitors Olaparib and Rucaparib received FDA approval for single-agent use in patients with mCRPC with HRR gene mutations based on results from the PROfound phase III clinical trial (NCT02987543). This trial demonstrated that Olaparib improved progression-free survival over second-generation hormone therapies in patients with mCRPC harbouring mutations in HRR genes including BRCA1/2 and ATM [18]. Consequently, the emergence of PARP inhibitors as anticancer drugs is a promising therapeutic option for men with mCRPC.

Currently, little is known about the impact of DDR inhibitors in combination with radiation strategies on mCRPC. Pre-clinical studies have identified ATM and ATR inhibitors as promising candidates when partnered with ^223^Ra [19,20]. Additionally, the phase I/II COMRADE clinical trial is investigating the synergistic activity of ^223^Ra in combination with Olaparib (NCT03317392) [21]. These studies highlight the potential of exploiting DDR inhibitors as a mechanism to increase the therapeutic efficacy of ^223^Ra.

In this study, we aimed to assess the in vitro radiobiological response of prostate cancer cell models to different radiation qualities (X-rays and ^223^Ra) in combination with inhibitors of key proteins of the DDR system, including an ATM inhibitor (AZD0156), an ATR inhibitor (AZD6738), and a PARP inhibitor (AZD2281). Moreover, we aimed to elucidate which drug-radiation combinations would be promising novel treatments for prostate cancer.

## 2. Materials and Methods

### 2.1. Cell Culture

Human prostate cancer cells (PC-3, DU145, and LNCaP) were purchased from the American Tissue Culture Collection (ATCC, Manassas, VI, USA). All cell lines were routinely cultured in RPMI-1640 supplemented with 10% FBS and 1% penicillin-streptomycin (Thermo Fisher (Waltham, MA, USA)). All cultures were incubated in a humidified atmosphere at 37 °C and routinely tested for mycoplasma.

### 2.2. Small-Molecule Inhibitors

The ATR inhibitor AZD6738 (AstraZeneca, Cambridge, UK), PARP inhibitor Olaparib (AZD2281), and ATM inhibitor (AZD0156) (Selleck Chemicals) were prepared in DMSO to a stock concentration of 10 mM and stored at −20 °C.

### 2.3. Radiation Treatments

X-ray irradiations (IR) from 0–8 Gy were delivered by an X-rad 225 irradiation system with a 2 mm copper filter (Precision X-RAY Inc., North Branford, CT, USA) at a dose rate of 0.57 Gy/min.

For exposure to radium-223 (^223^Ra), Xofigo (Bayer, Bayer Leverkusen, Germany) with activities ranging from 0.5 MBq/mL to 0.1 MBq/mL were kindly donated by the Northern Ireland Cancer Centre. Microdosimetry calculations were made to determine the ^223^Ra volume to be added to each well based on the activity of the vial, exposure time, and desired absorbed dose. Cells were exposed to doses ranging from 0 to 0.5 Gy in up to 50 μL of Xofigo added to 2 mL of cell culture medium in a 6-well plate, with an exposure time of 24 h. To ensure the same volume was added to all conditions independently of dose rate or exposure time, ^223^Ra solution was previously diluted in saline solution (0.1 M sodium chloride, 0.02 M sodium citrate, and 5 mM hydrochloric acid (all reagents from Sigma-Aldrich, Glasgow, UK), at pH 7.0), ensuring the total volume was equal to the Xofigo solution added to the highest treatment. Based on all alpha particle decays from the ^223^Ra cascade, the average emission energy is 6.67 MeV, and the average entrance LET is 72 keV/μm. For all experiments, control samples were treated with a saline solution containing the same compounds as the Xofigo solution, except for ^223^Ra. After 24 h exposure, the treatment medium was removed, and cells were washed 3 times in PBS. Cells were then incubated at 37 °C with fresh, complete medium before analysis. Additional details on ^223^Ra in vitro treatment and dose calculations are described in Liberal et al., 2022 [22].

### 2.4. Clonogenic Survival Assay

Colony formation assays were carried out according to published methods [23]. Briefly, cells were plated into six-well plates (Sarstedt AG & Co., Nümbrecht, Germany) with an optimal cell density for colony formation depending on the cell line and absorbed dose. After 24 h, cells were irradiated with either 0–8 Gy X-rays or exposed to a range of activities of ^223^Ra for 24 h to deliver radiation doses from 0–0.5 Gy.

For IC_50_ clonogenics, cells were seeded at an optimal density and incubated. After 24 h cells were treated with a dose range of 0.01 µM to 5 µM of AZD0156, AZD6738, and AZD2281.

For assessment of small-molecule inhibitors, cells were pre-treated with non-toxic concentrations of AZD6738 (100 nM), AZD0156 (100 nM), and AZD2281 (500 nM), based on IC_50_ values, or DMSO for 1 h before exposure to equivalent isotoxic doses of radiation qualities (2 Gy dose of X-rays or 0.25 Gy dose of ^223^Ra).

For all clonogenic experiments, cells were incubated for 7–10 days to allow for colony formation. Colonies were fixed and stained with 2% crystal violet in 80% methanol. Colonies greater than 50 cells were scored as representing surviving cells. The plating efficiency percentage (PE) was determined by calculating the number of counted colonies divided by the number of cells seeded times 100%. The survival fraction was determined by the number of colonies formed after treatment divided by the number of cells seeded, corrected for the PE of unirradiated cells.

Data was fitted to a linear quadratic model using non-linear regression of the form SF = exp[−(αD+ βD^2^)], where SF = survival fraction, D = dose, and α and β are constants. Statistical errors on fit values were calculated as the standard error. Relative biological effectiveness (RBE) for SF = 50% was calculated to compare ^223^Ra to X-rays for all cell lines examined in the current study. For all treatment groups, combination indices (CI) were calculated [24]. CI < 0.9 indicated a synergistic interaction.

### 2.5. DNA Damage Assay

Cells were seeded on coverslips at a density of 1 × 10^6^ per well on 18 × 18 mm autoclaved coverslips (Mensel Glaser, Braunschweig, Germany) and allowed to adhere for 24 h. Following this, cells were pre-treated with concentrations of AZD6738 (100 nM), AZD0156 (100 nM), and AZD2281 (500 nM) and exposed to isotoxic doses of low and high LET radiation as described previously. Following a 1 h or 24 h incubation period, 50% (*v*/*v*) ice-cold methanol-acetone was used to fix cells, and cells were then permeabilised using 0.5% Triton X-100 before being blocked in blocking buffer (5% FBS and 0.1% Triton X-100 in PBS). Next, cells were incubated in anti53BP1 (NB100-304, Novus Biologicals, Centennial, CO, USA) diluted in blocking buffer (1:5000) at 4 °C for 1 h. Following washing, Alexaflour 568 goat anti-rabbit IgG secondary antibody (A21429, Invitrogen (Waltham, MA, USA)) diluted in blocking buffer (1:2000) was added for 1 h at room temperature under dark conditions. After staining, cells were washed in PBS 3 times and mounted using Prolong Gold antifade reagent with DAPI (P36930, Invitrogen, Waltham, MA, USA) onto microscope slides. Foci were scored using a Zeiss Axiovert 200 M microscope (Carl Zeiss MicroImaging, LLC, White Plains, NY, USA) at a magnification of ×63. For each treatment condition, 53BP1 foci were determined in at least 50 cells. Data are presented as the mean number of foci per cell ± SD of three independent repeats.

### 2.6. Flow Cytometry Analysis of Cell Cycle

Cells were pre-treated with concentrations of AZD6738 (100 nM), AZD0156 (100 nM), and AZD2281 (500 nM) and exposed to isotoxic doses of radiation qualities as described previously. Cells were harvested at 24 h and fixed using 100% ice-cold ethanol. Samples were resuspended in 360 μL of PI/RNaseA and incubated for 30 min at 37 °C before being analysed by flow cytometry on a BD Acuri C6 Plus Flow Cytometer (San Jose, CA, USA). BD Accuri C9 Plus Analysis software version 1.0.23 was used for quantification.

### 2.7. Western Blotting

Prior to radiation, cells were treated with concentrations of AZD6738 (100 nM), AZD0156 (100 nM), and AZD2281 (500 nM) and exposed to isotoxic doses of different quality radiations as described previously. Cells were harvested at 48 h and extracted according to published methods [25]. Forty µg of each sample was loaded onto Invitrogen NuPAGE 8% Bis-Tris Midi gels. After electrophoresis, proteins were transferred onto Invitrogen IBlot2 regular stacks using an IBlot. The membranes were blocked and incubated with primary antibodies [PARP [#9542] (Cell Signalling, Danvers, MA, USA)] at a 1:1000 dilution at 4 °C overnight. β-actin (Cell Signaling, USA) was used as a loading control at a 1:5000 dilution. After washing with PBS-T, membranes were incubated at room temperature for 1 h in their secondary anti-rabbit and anti-mouse horseradish peroxidase-conjugated antibodies at a 1:2000 dilution. The membranes were then washed and developed by ECL reagent (7.5 mL Tris HCl, 16.5 µL coumaric acid, 37.5 µL luminor, 2.5 µL H_2_O_2_) using the GBox Imager by Syngene (Cambridge, UK).

### 2.8. Statistical Analysis

All experiments were performed in triplicate. Unpaired Student’s *t*-test and one-way ANOVA were used for statistical evaluation. All calculations were performed using GraphPad Prism 7.0 (GraphPad, Boston, MA, USA). Probability values were classified as **** (*p* < 0. 0001), *** (*p* < 0. 001), ** (*p* < 0.01), and * (*p* < 0.05).

## 3. Results

### 3.1. The Radiobiological Response of Prostate Cancer Cells to X-rays and ^223^Ra

Clonogenic assays were performed on human prostate cancer cell lines to elucidate the effect of different radiation qualities on cell survival. Radiobiological parameters for each of the survival curves in Figure 1 are presented in Table 1. ^223^Ra treatment radiosensitised all cell lines examined relative to X-rays (Figure 1A,B). For all ^223^Ra treatments, there was a greater than 10-fold increase in the α values and a decrease in the β values, with the exception of the PC-3 cells, as expected from these high LET exposures. All cell types were highly sensitive to radium treatment, with RBE values at 50% survival varying from 9.91 ± 0.5 for PC-3 cells to 21.62 ± 0.6 for DU145 cells.

Based on survival fraction (SF) at doses of X-rays (2 Gy) and ^223^Ra (0.25 Gy), the DU145 cell line was the most radioresistant cell line, with a SF of 0.68 ± 0.02 after X-rays and 0.42 ± 0.02 after ^223^Ra. In comparison, the PC-3 cell line was the most radiosensitive cell line to both radiations, with a SF of 0.43 ± 0.04 and 0.33 ± 0.02, to X-rays and ^223^Ra, respectively.

### 3.2. The Effect of DDR Inhibition on Cell Survival in Prostate Cancer Cell Lines

The IC_50_ values of AZD0156, AZD6738, and AZD2281 for PC-3, DU145, and LNCaP prostate cancer cell lines were determined using clonogenic assays (Appendix A). All DDR inhibitors reduced clonogenic survival in a dose-dependent manner across all tested cell lines. AZD0156 was the most potent DDR inhibitor, with IC_50_ values of 0.28 µM, 0.30 µM, and 0.29 µM in PC-3, DU145, and LNCaP cell lines, respectively. In comparison, AZD6738 IC_50_ values of 0.43 µM, 0.43 µM, and 0.38 µM in PC-3, DU145, and LNCaP cell lines, respectively, were determined. AZD2281 was the least potent of all the DDR inhibitors examined, with IC_50_ values of 4.24 µM, 27.44 µM, and 50.64 µM in PC-3, DU145, and LNCaP cell lines, respectively.

For combination studies, concentrations of 100 nM were chosen for AZD0156, 100 nM for AZD6738, and 500 nM for AZD2281 across all cell lines, as we wanted to use non-toxic concentrations of the DDR inhibitors in order to clearly detect combined inhibitor radiation-induced effects.

### 3.3. DDR Inhibitors Potentiate Radiosensitisation in Prostate Cancer Cell Lines Treated with X-rays and ^223^Ra

Clonogenic assays were employed to assess doses of radiation of different qualities (2 Gy X-rays or 0.25 Gy ^223^Ra) alone or in combination with DDR inhibitors on prostate cancer cell lines (Figure 2). These doses were selected to take into account the increased RBE of α-particle exposure relative to X-rays. Additionally, Table 2 presents Combination Index (CI) values for each combinatory treatment in each cell line; CI values equal to or inferior to 0.9 were considered synergistic.

Treatment with either X-rays or ^223^Ra alone significantly reduced cell survival across all cell lines in comparison to untreated controls (*p* < 0.001). A comparison between radiation qualities determined that ^223^Ra was a more potent radiosensitiser in all cell lines examined in comparison to X-rays (PC-3 and DU145 cells *p* ˂ 0.01; LNCaP cells *p* ˂ 0.05).

For all DDR inhibitors assessed in combination with either X-rays or ^223^Ra, a synergistic interaction was determined, with the greatest radiosensitisation response observed after ^223^Ra in combination with DDR inhibitors in comparison to X-rays and DDR inhibitor combinations (*p* ˂ 0.05). This is because each CI value of each inhibitor combined with ^223^Ra was remarkably lower than when combined with X-rays (e.g., AZD0516 CI value of 0.84 ± 0.19 when combined with X-rays vs. CI of 0.043 ± 0.32 in the PC-3 cell model). Moreover, the strongest synergistic combination with X-rays across all cell lines was the ATR inhibitor AZD6738 (CI = PC-3 0.73 ± 0.11 (*p* ˂ 0.01); DU145 0.38 ± 0.18 (*p* ˂ 0.0001); LNCaP 0.45 ± 0.17 (*p* ˂ 0.05)) (Figure 2D–F). AZD6738 and AZD0156 were both potent radiosensitisers when combined with ^223^Ra across all cell lines (AZD6738 CI values = PC-3 0.16 ± 0.54 (*p* ˂ 0.001); DU145 0.20 ± 0.14 (*p* ˂ 0.001); LNCaP 0.07 ± 0.32 (*p* ˂ 0.001)) (AZD0156 CI values = PC-3 0.04 ± 0.32 (*p* ˂ 0.0001); DU145 0.41 ± 0.11 (*p* ˂ 0.01); LNCaP 0.12 ± 0.09 (*p* ˂ 0.001)). In comparison, a less synergistic interaction was determined with AZD2281 in combination with ^223^Ra in DU145 and LNCaP cells (CI = 0.56 ± 0.13 (*p* ˂ 0.01) and 0.67 ± 0.03 (*p* ˂ 0.01), respectively).

### 3.4. DDR Inhibitor-^223^Ra Combinations Mediate Rate of DNA DSB Repair

Immunofluorescence detection of 53BP1, a DNA damage response protein recruited to DSB sites, was used to quantify the effects of different radiation qualities alone and in combination with DDR inhibitors on DNA DSB repair in PC-3, DU145, and LNCaP prostate cancer cell lines (Figure 3). Representative images of foci are presented in Appendix A.

Exposure to either X-rays (2 Gy) or ^223^Ra (0.25 Gy) alone resulted in significantly higher levels of 53BP1 1 h after treatment (PC-3 foci 26 vs. 25.7; DU145 foci 21.3 vs. 19.3; LnCAP foci 20.5 vs. 18.1, respectively), with DSBs repairing in a cell- and radiation quality-dependent manner over 24 h. Across all cell lines, no significant difference in the induction of foci at 1 h was determined between the different radiation qualities examined. However, in comparison to X-rays, the rate of DSB repair after ^223^Ra exposure was slower as 53BP1 foci levels were significantly higher at 24 h across all cell lines (PC-3 foci 8.18 vs. 12.4; DU145 foci 2.96 vs. 6.81; LnCAP foci 4.43 vs. 8.42, respectively) (PC-3 and DU145 cells *p* ˂ 0.01; LNCaP cells *p* ˂ 0.001).

Treatment with DDR inhibitors alone had no significant effect on 53BP1 foci in PC-3 cells. In comparison, AZD0156 and AZD6738 significantly increased the number of 53BP1 foci in DU145 and LNCaP cells (*p* ˂ 0.05), whereas AZD2281 only significantly increased mean levels of 53BP1 foci in DU145 cells (*p* ˂ 0.01). At 1 h after combined X-rays and DDR inhibitor treatment, PC-3 cells exhibited significantly greater 53BP1 foci in comparison to X-rays alone across all inhibitors assessed (*p* ˂ 0.05). In comparison, only AZD2281 in combination with X-rays significantly increased initial 53BP1 foci numbers at 1 h in DU145 cells (*p* ˂ 0.05). In contrast, ^223^Ra in combination with AZD0156 or AZD6738 significantly increased 53BP1 foci at 1 h in both PC-3 and LNCaP cells (*p* ˂ 0.05). However, AZD6738, in combination with ^223^Ra, was the only inhibitor to significantly increase initial levels of 53BP1 foci 1 h after treatment exposure in DU145 cells (*p* ˂ 0.05). Following 24 h, the number of 53BP1 foci after X-rays in combination with DDR inhibitors was significantly greater across all cell lines (*p* ˂ 0.05) apart from PC-3 cells, where no significant difference in foci numbers was determined after X-rays in combination with AZD0156. In contrast, for all DDR inhibitors examined in combination with ^223^Ra, 53BP1 foci levels remained significantly elevated 24 h after treatment in comparison to ^223^Ra exposure alone (*p* ˂ 0.05). Moreover, a comparison between radiation qualities with DDR inhibitors identified that, apart from DU145 cells treated with AZD2281 and AZD6738, 53BP1 foci levels remain significantly elevated after ^223^Ra treatment in comparison to cells treated with X-rays at 24 h (*p* ˂ 0.01). In general, of all the DDR inhibitors assessed, AZD6738, in combination with both radiation qualities, induced the greatest number of 53BP1 foci at 24 h.

### 3.5. Combined DDR Inhibitors and Radiation Qualities on Cell Cycle Distribution

Previous results have shown higher levels of residual damage 24 h after combined treatment of DDR inhibitors with radiation of different qualities. These increased levels of residual damage are a direct result of a slower and/or less efficient repair of radiation-induced DNA DSB. The cell cycle distribution was therefore studied to determine whether DNA damage caused changes in the cell cycle distribution. Cell cycle profiles of PC-3, DU145, and LNCaP prostate cancer cell lines were determined by flow cytometry using propidium iodine (PI) staining of DNA (Figure 4).

No significant changes in cell cycle distribution in any of the prostate cancer cells examined were determined 24 h after 2 Gy X-rays or treatment with DDR inhibitors alone in comparison to untreated samples. Furthermore, under these conditions and at a 24 h timepoint, no significant sub-G1 was detected. However, across all 3 cell lines, ^223^Ra significantly reduced the proportion of cells in the G1 phase (PC-3 *p* < 0.05; DU145 *p* < 0.001; LNCaP *p* < 0.01). Furthermore, a significant increase in the proportion of cells in the G2/M phase was determined in DU145 and LNCaP cells treated with 0.25 Gy ^223^Ra in comparison to untreated controls (*p* < 0.0001). Therefore, it is evident that high LET radiation, such as ^223^Ra, had a more profound impact on the cell cycle distribution in comparison to X-rays, as demonstrated by the more efficient and prolonged G2 arrest in DU145 and LNCaP cells (*p* < 0.05 and *p* < 0.01, respectively).

Combined X-rays and AZD0156 led to a significant accumulation of DU145 and LNCaP cells in the G2/M phase in comparison to treatment with X-rays alone (*p* < 0.001), with no significant changes identified in the PC-3 cell cycle profile. Similarly, combining AZD0156 with ^223^Ra also led to a significant increase in the G2/M population and a reduction of cells in the G1 phase of the cell cycle across all three cell lines in comparison to ^223^Ra-treated samples (*p* < 0.05). Also, significant G2/M arrest after combined AZD2281 and X-rays was only determined in DU145 and LNCaP cells (*p* < 0.05), whereas ^223^Ra, when combined with AZD2281, caused a significant accumulation of cells in the G2/M phase of DU145 and PC-3 cells (*p* < 0.01 and *p* < 0.0001, respectively). A comparison between these two DDR inhibitors determined that AZD0156 compared with AZD2281 in combination with both radiation qualities resulted in the greatest percentage of cells in the G2/M phase across all three cell lines. For instance, the proportion of cells in G2/M for DU145 cells was 38% ± 1.1 (X-rays + AZD0156) vs. 35% ± 2.10 (X-rays + AZD2281) and 51% ± 0.06 (^223^Ra + AZD0156) vs. 45% ± 1.8 (^223^Ra + AZD2281).

In contrast, AZD6738 in combination with X-rays abrogated the G2/M cell cycle checkpoint, with a significant reduction in the percentage of cells in G2/M (*p* < 0.01) and an increase in the accumulation of cells in the G1 phase for PC-3 cells (*p* < 0.001). Similarly, ^223^Ra and AZD6738 also abrogated radiation-induced G2 arrest across all cell lines examined (*p* < 0.05).

### 3.6. DDR-Inhibitor-^223^Ra Combinations Trigger Apoptosis Induction

Expression of PARP-1 cleavage in human PC-3 prostate cancer cells was used as a surrogate marker of apoptotic cell death 48 h after radiation exposure of different qualities alone or in combination with DDR inhibitors. In comparison to X-rays, ^223^Ra induced slightly greater expression levels of cleaved PARP-1 (Figure 5). Although no differences were determined in the expression of PARP-1 cleavage after the combination of DDR inhibitors and X-rays, higher levels of PARP-1 cleavage were observed after the combination of DDR inhibition with ^223^Ra. Whole western blots including molecular weight markers are presented in Appendix A.

## 4. Discussion

Retrospective studies have determined a greater clinical benefit following ^223^Ra treatment for patients with mCRPC harbouring deficiencies in HRR-mediated DNA repair genes [26,27]. In the present study, we aimed to elucidate whether DDR inhibitors had the ability to radiosensitise prostate cancer cell lines to X-rays or ^223^Ra in vitro and to better understand the relationship between the combined effects of radiation qualities with DDR inhibitors on radiobiological response.

Across the prostate cancer cell lines assessed in this study, ^223^Ra was the most effective at cell killing in comparison to X-rays as determined by RBE_50%_ values (Figure 1 and Table 1). In general, ^223^Ra was ~10 times more effective than the equivalent dose of X-rays. This is consistent with previously reported pre-clinical studies that compared radio-response with different radiation qualities [22,28]. Also, in good agreement with previous results, we observed individual differences in cellular response to radiation regardless of its quality, which may be attributed to differentially expressed genes amongst cell lines [28].

The therapeutic landscape of anti-tumour agents targeting the DDR has rapidly evolved, with multiple DDR inhibitors emerging in the clinic for a range of cancers. ATR inhibitors have gained significant interest recently due to their promising potential in preclinical trials. Currently, ATR inhibitors in clinical development include VX-970, AZD6738, and BAY1895344. In localised and mCRPC, these inhibitors are being assessed as monotherapies and in combination with PARP inhibitors, radiation therapies, and chemotherapies [29]. As a monotherapy, ATR inhibitors have demonstrated synergistic efficacy in ATM-deficient prostate cancer models [13]. Moreover, as ATR and PARP function in independent DDR pathways, concomitant inhibition of both proteins has been shown to enhance antitumor activity, with a superior efficacy observed in ATM-knockout prostate models [30,31]. Similarly, for PARP inhibitors, much of the focus in prostate cancer has been on the “synthetic lethality” mechanism, which has resulted in these agents becoming available for the treatment of patients with BRCA1/2 or ATM-loss prostate cancer [18]. In comparison, the efficacy of ATM inhibitors for the treatment of prostate cancer has not been extensively studied; however, there are three ATM inhibitors (M354, AZD0516, and AZD1390) undergoing clinical investigation.

Consistent with previously published data, we determined that the ATR inhibitor, AZD6738, and the PARP inhibitor, AZD2281, radiosensitised all prostate cancer cells to low LET (Figure 2 and Table 2) [32,33]. Interestingly, the ATM inhibitor AZD0156 failed to radiosensitise PC-3 cells to X-rays. This observation is consistent with previously reported data using the same cell line [34]. In contrast, we identified that ATM inhibition in combination with X-rays significantly reduced the cell survival of DU145 and LNCaP cells. A similar response for DU145 but not LNCaP cells was previously reported [34]. We speculate that the differences in data may be explained by the small-molecule inhibitor investigated, as the ATM inhibitor KU-60019 was reportedly used by Hanna et al., 2021 [34].

Pre-clinical investigations comparing the efficacy of DDR inhibitors in combination with ^223^Ra, or other high-LET radiations, in prostate cancer are limited. In agreement with previous studies in different cancer models that combined ATM or PARP inhibitors with either proton or carbon ion irradiation, we determined that pharmacological inhibition of these pathways attenuated cell survival in combination with high LET ^223^Ra radiation [35,36]. Additionally, retrospective studies have demonstrated greater clinical benefit from ^223^Ra for patients harbouring DNA repair mutations, including ATM mutations [26,27]. In line with the results presented here, previous studies have also demonstrated the synergistic activity of the ATR inhibitor BAY1895344 in combination with ^223^Ra across a range of in vitro and in vivo prostate cancer models [20,33]. Moreover, Fujisawa et al. (2015) reported the efficacy of the ATR inhibitor VE-821 with low and high LET radiation sources in human tumour cells, indicating inhibition of this protein as an attractive therapeutic strategy in combination with differing radiation qualities [37]. From this research study, a comparison between all three DDR inhibitors identified that the ATR inhibitor, AZD6738, selectively enhanced the relative biological effectiveness of both low and high LET sources examined in comparison to AZD0156 and AZD2281. One possible explanation for this may be due to cell survival dependency on the ATR kinase, which is prevented when an inhibitor targeting this pathway is present [38].

The inability of cells to repair DSBs after ^223^Ra exposure is widely reported to be associated with the dense ionisation of α-particles and their characteristic high LET, leading to complex lesions that are difficult to repair and ultimately inducing cell death [39,40]. This is supported by our observations of elevated levels of unrepaired DNA damage 24 h after ^223^Ra treatment (Figure 3). This was associated with an increase in PARP-1 cleavage, which is a surrogate marker of apoptosis (Figure 5). In contrast, the DDR system had the ability to alleviate X-ray-induced DSBs at 24 h, with no expression of cleaved PARP-1 observed. These findings indicate the superior efficacy of ^223^Ra to induce complex clusters of irreparable DSBs.

In our study, DDR inhibitors in combination with radiation qualities resulted in an increased number of unrepaired 53BP1 foci at 24 h, with the greatest DNA damage observed after DDR inhibitors were combined with ^223^Ra in comparison to X-rays. Furthermore, combinations of DDR inhibitors and ^223^Ra resulted in the greatest levels of apoptosis. To the best of our knowledge, this is the first study to assess DSB repair in response to high-LET radiation combined with DDR inhibitors in prostate cancer cells. For PC-3 cells, the combination of ATM inhibition and X-rays had no significant impact on levels of residual damage in comparison to X-rays alone. These results mirror clonogenic survival data, which determined no enhanced radiobiological effect of this particular combination in PC-3 cells. For both radiation qualities assessed, significantly more 53BP1 foci persisted at 24 h across all human prostate cancer cell lines pre-treated with AZD678, indicating persistent unrepaired DNA damage. For ^223^Ra, inhibition of ATM also resulted in elevated levels of 53BP1. Interestingly, although significant, PARP inhibition resulted in the least 53BP1 accumulation in combination with ^223^Ra in comparison to the other DDR inhibitors assessed. Based on the literature, it is possible that the efficacy of the PARP inhibitor may be dependent on the presence of additional HRR mutations, such as ATM and BRCA1/2 mutations; however, further studies are warranted to validate this.

In response to radiation exposure and to allow time for activation of DNA repair mechanisms, cell cycle checkpoints regulate the arrest or progression of the cell cycle. Cells that have undergone radiation-induced DNA damage predominantly rely on G2/M checkpoints for arrest and repair of DSBs [41]. Cells that fail to properly activate G2 arrest, ultimately entering mitosis, will undergo mitotic catastrophe due to partially replicated sister chromatids condensing and causing DNA to shatter [42]. In agreement with previous studies that examined cell cycle responses to high and low radiation exposures, our cell cycle analyses showed that changes in cell cycle distribution were dependent on the type of radiation exposure (Figure 4) [19,22,43,44]. Low LET X-rays increased the G2/M population across all cell lines, albeit not significantly. In contrast, ^223^Ra exposure increased G2/M arrest and was accompanied by a resultant decrease in the G1 phase in cells, which was significant in DU145 and LNCaP cells. The differences in cell cycle distribution between low and high LET radiation are a consequence attributed to DNA DSB severity. Cells exposed to X-rays have the ability to quickly repair DSB damage and continue through the cell cycle, whereas complex DSB damage caused by ^223^Ra is more difficult for cells to repair, and ultimately the cells remain in the G2/M phase of the cell cycle [22,43,44].

In the present study, the number of cells arrested in the G2/M phase marginally increased, although not significantly, across the entire panel of prostate cell lines after ATM inhibition and PARP inhibition-radiation quality combinations, consistent with published data [19,32]. Furthermore, the effects exhibited by these inhibitors in combination with ^223^Ra on G2/M cell cycle arrest were more pronounced than those displayed by X-rays combined with ATM and PARP inhibition. In comparison, ATR-radiation quality combinations decreased the percentage of G2/M phase, indicating abrogation of G2/M cell cycle arrest, which has also been reported by several previous studies [37,45].

## 5. Conclusions

The clinical application of targeted α- and β-radionuclide therapies for prostate cancer is rapidly evolving with the development of novel next-generation radiopharmaceutical treatments, including ^177^Lu-PSMA, which target bone and extraskeletal metastases, gaining significant attraction as potential therapeutic options for patients with mCRPC. Targeting the DDR pathways through inhibition of specific kinases remains an area of interest as a therapeutic strategy to enhance anti-tumour efficacy. In this study, we assessed the radiobiological effects of X-rays and ^223^Ra in combination with several DDR inhibitors. Our findings highlight the potential of exploiting DDR inhibitors, particularly AZD6738, as a mechanism to increase the therapeutic efficacy of X-rays and ^223^Ra in in vitro prostate cancer cell lines. Further pre-clinical studies are required to investigate changes in molecular mechanisms co-occurring with DDR inhibitors in combination with low and high LET radiation. Additionally, further studies should be carried out to confirm apoptosis as a mechanism of cell death for these combinations, including measurements of caspases 3, 8, and 9. To extend the concepts outlined in this paper, in vivo evaluation should be undertaken to provide proof-of-concept for the application of DDR inhibitors in combination with ^223^Ra in future phase 1 clinical trials for men with mCRPC. Future studies should include assessing caspases 3 and 9 to confirm apoptosis.

## Figures and Tables

**Figure 1 cancers-16-01510-f001:**
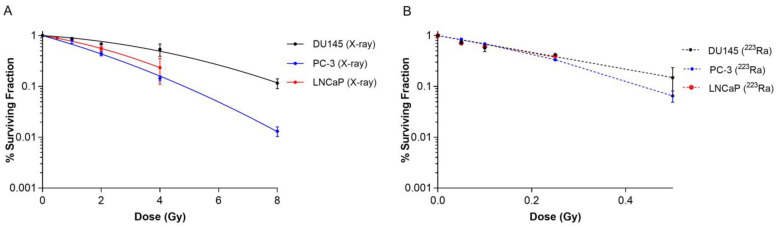
Survival curves for PC-3, DU145, and LNCaP human prostate cancer cell lines after exposure to different doses of X-rays (0–8 Gy) (**A**) or ^223^Ra (0–0.5 Gy) (**B**) for 24 h. After 7–10 days of culture, colonies were counted and survival fractions calculated. The data were fit to a linear quadratic model. Points represent the mean from three independent experiments and the respective standard error.

**Figure 2 cancers-16-01510-f002:**
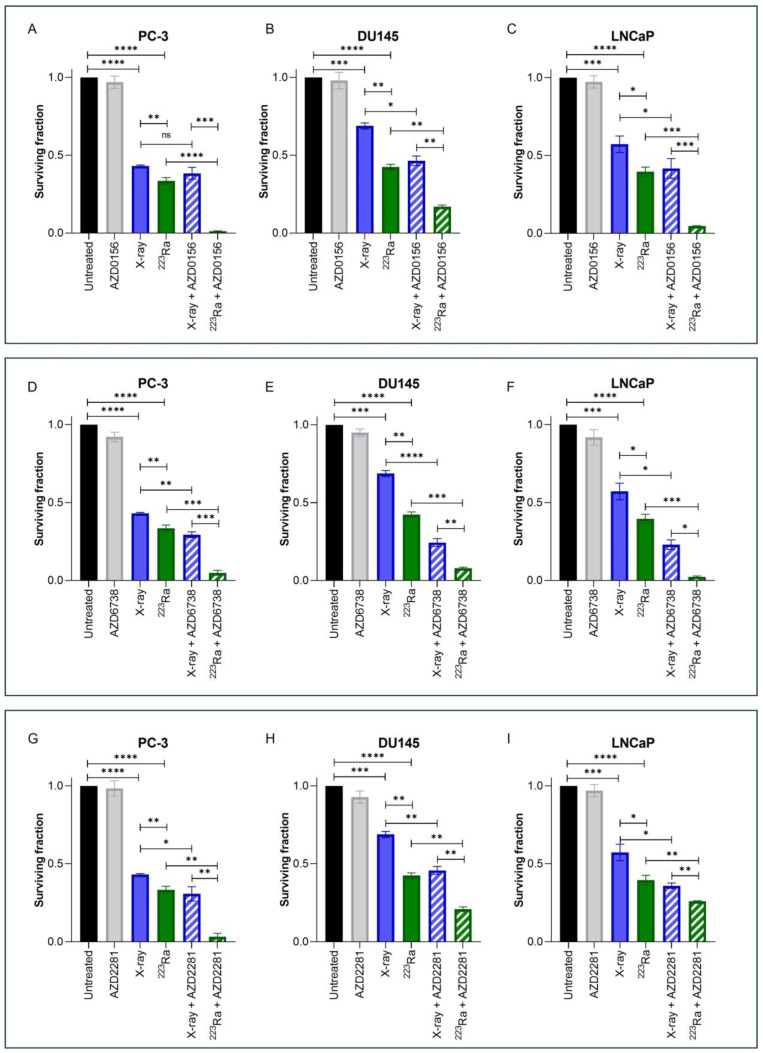
Clonogenic survival response for PC-3 (**A**,**D**,**G**), DU145 (**B**,**E**,**H**), and LNCaP (**C**,**F**,**I**) human prostate cancer cell lines to isotoxic doses of X-rays (2 Gy) or ^223^Ra (0.25 Gy) alone or in combination with AZD0156 (100 nM) (**A**–**C**), AZD6738 (100 nM) (**D**–**F**), or AZD2281 (500 nM) (**G**–**I**). After 7–10 days of culture, colonies were counted and survival fractions calculated. Each value represents the mean from three independent experiments and the respective standard error. Differences between two groups were compared by using an unpaired Student’s *t*-test (**** *p* < 0. 0001, *** *p* < 0. 001, ** *p* < 0.01, * *p* < 0.05, non-significant (ns)).

**Figure 3 cancers-16-01510-f003:**
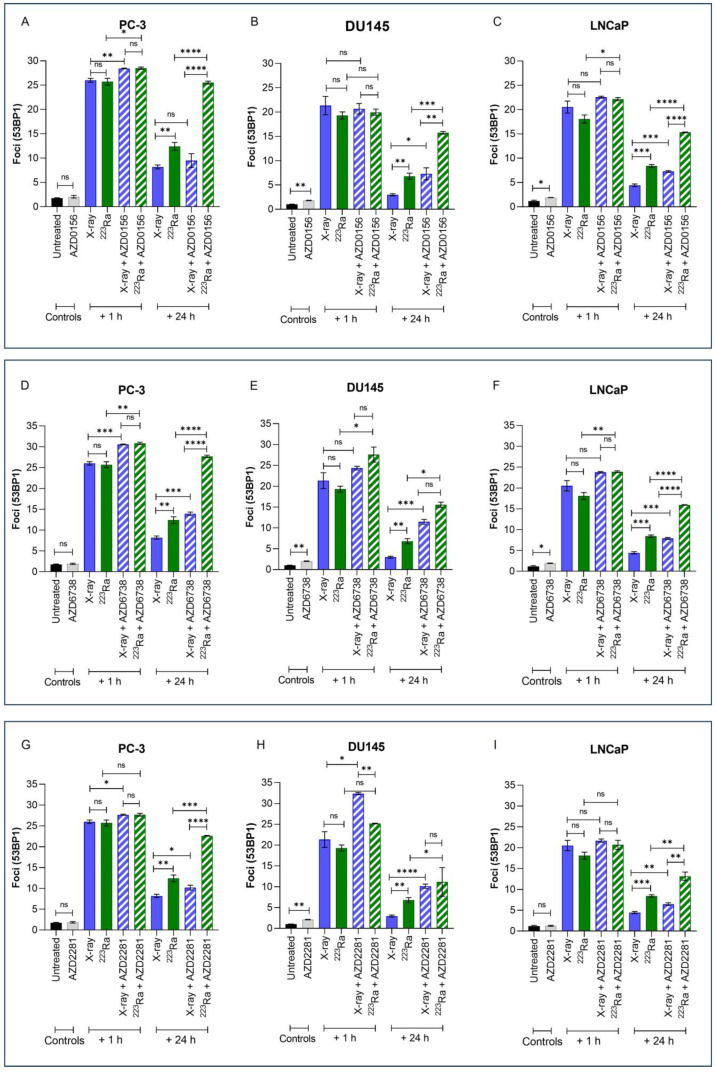
Mean 53BP1 foci per cell for PC-3 (**A**,**D**,**G**), DU145 (**B**,**E**,**H**), and LNCaP (**C**,**F**,**I**) human prostate cancer cell lines, corrected for background levels, was plotted at 1 h and 24 h following X-rays (2 Gy) or ^223^Ra (0.25 Gy) alone or in combination with AZD0156 (100 nM) (**A**–**C**), AZD6738 (100 nM) (**D**–**F**), or AZD2281 (500 nM) (**G**–**I**). Each value represents the mean from three independent experiments and the respective standard error. Differences between two groups were compared by using an unpaired Student’s *t*-test (**** *p* < 0. 0001, *** *p* < 0. 001, ** *p* < 0.01, * *p* < 0.05, non-significant (ns)).

**Figure 4 cancers-16-01510-f004:**
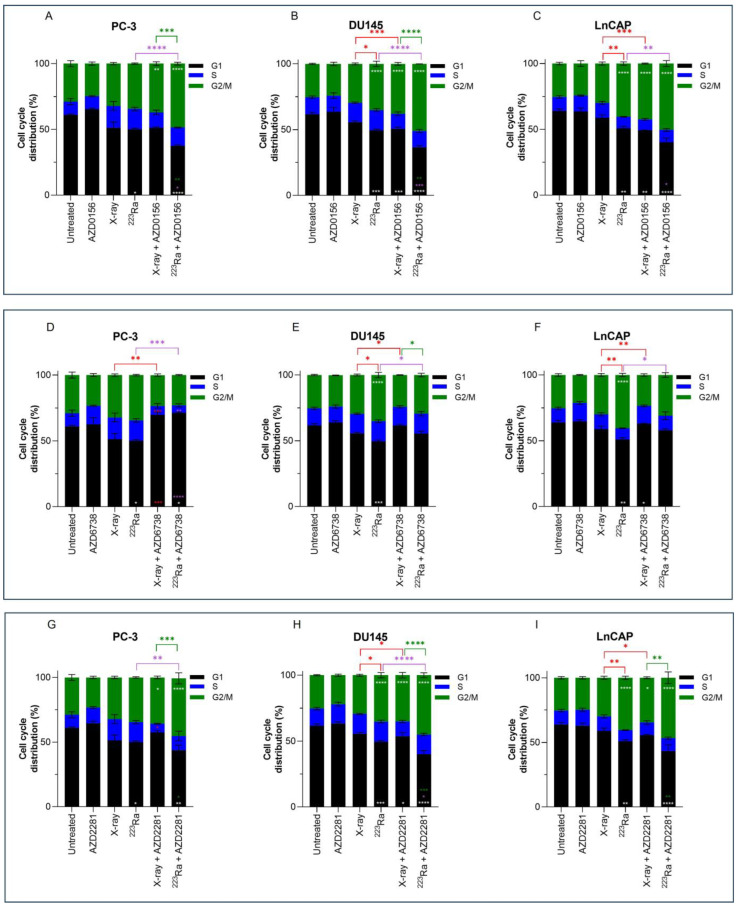
Cell cycle analysis of PC-3 (**A**,**D**,**G**), DU145 (**B**,**E**,**H**), and LNCaP (**C**,**F**,**I**) human prostate cancer cell lines 24 h after treatment with X-rays (2 Gy) or ^223^Ra (0.25 Gy) alone or in combination with AZD0156 (100 nM) (**A**–**C**), AZD6738 (100 nM) (**D**–**F**), or AZD2281 (500 nM) (**G**–**I**). Bars represent the mean from three independent experiments and the respective standard error. Significant differences for control vs. all groups (white stars), X-rays vs. all groups (red stars), ^223^Ra vs. all groups (purple stars), and X-rays + DDR inhibitors vs. ^223^Ra vs. DDR inhibitors (green stars) are represented by (**** *p* < 0. 0001, *** *p* < 0. 001, ** *p* < 0.01, * *p* < 0.05).

**Figure 5 cancers-16-01510-f005:**
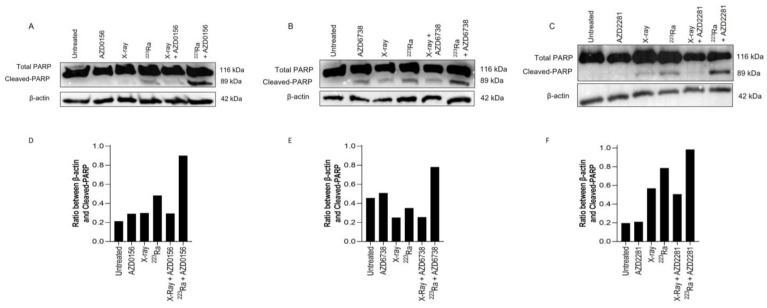
Western blot for protein expression of full and cleaved PARP-1 in the PC-3 human prostate cancer cell line at 48 h after treatment with X-rays (2 Gy), ^223^Ra (0.25 Gy), AZD0156 (100 nM) (**A**), AZD6738 (100 nM) (**B**), or AZD2281 (500 nM) (**C**) alone or in combination. β-actin was used as a loading control. (**D**–**F**) The intensity ratio between Cleaved-PARP and β-actin.

**Table 1 cancers-16-01510-t001:** Linear quadratic parameters for PC-3, DU145, and LNCaP prostate cancer cell lines after exposure to different radiation qualities.

Parameters		PC-3	DU145	LNCaP
X-rays	α (Gy^−1^)	0.37 ± 0.1	0.09 ± 0.09	0.2 ± 0.04
	β (Gy^−2^)	0.02 ± 0.02	0.02 ± 0.01	0.04 ± 0.01
^223^Ra	α (Gy^−1^)	3.33 ± 0.1	3.8 ± 0.6	3.93 ± 1.58
	β (Gy^−2^)	4.27 ± 0.3	~0	~0
RBE_50%_ (X-ray/^223^Ra)		9.91 ± 0.5	21.62 ± 0.6	13.01 ± 0.1

**Table 2 cancers-16-01510-t002:** Combination indices with respective standard errors for PC-3, DU145, and LNCaP human prostate cancer cell lines after treatment with isotoxic doses of X-rays (2 Gy) or ^223^Ra (0.25 Gy) alone or in combination with AZD0156 (100 nM), AZD6738 (100 nM), or AZD2281 (500 nM).

Cell Line	Treatment Group					
	X-ray + AZD0516	X-ray + AZD6738	X-ray + AZD2281	^223^Ra + AZD0156	^223^Ra + AZD6738	^223^Ra + AZD2281
PC-3	0.84 ± 0.19	0.73 ± 0.11	0.73 ± 0.26	0.04 ± 0.32	0.16 ± 0.54	0.10 ± 0.90
DU145	0.70 ± 0.10	0.38 ± 0.18	0.56 ± 0.13	0.41 ± 0.11	0.20 ± 0.14	0.56 ± 0.13
LNCaP	0.71 ± 0.21	0.45 ± 0.17	0.69 ± 0.13	0.12 ± 0.09	0.07 ± 0.32	0.67 ± 0.03

## Data Availability

The original contributions presented in the study are included in the article/Appendix A. Further inquiries can be directed to the corresponding author.

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
