# Peer review of "Synergistic Activity of DNA Damage Response Inhibitors in Combination with Radium-223 in Prostate Cancer"

_cancers, 2024, doi:10.3390/cancers16081510_

Round 1

Reviewer 1 Report

Comments and Suggestions for Authors

This is a very nice article, very nice work by the author. I only have a couple of comments:

In the introduction you are mentioning DSB and complex DSB. In your results section you did not present any amount of DSB. Do you have this type of data? Will it be possible to include this?

L137-140: Check that you need this sentence here, as you have already mentioned this in line 129. Maybe you did that for completeness, but just make sure.

Author Response

We thank the reviewer for taking the time to read the manuscript and provide comment's. Please see below in bold, responses to comments. 

In the introduction you are mentioning DSB and complex DSB. In your results section you did not present any amount of DSB. Do you have this type of data? Will it be possible to include this? Additional detail has been added at lines 290-291 and 296-297 giving numbers of absolute foci at 1 and 24 h. We don’t have specific quantification of complex DSBs but given the difference in residual damage in X-rays relative to 223Ra, it is likely that 223Ra, given its high LET induces more complex DSBs.

L137-140: Check that you need this sentence here, as you have already mentioned this in line 129. Maybe you did that for completeness, but just make sure. Line 138-140 has been removed as this was repetition.

Reviewer 2 Report

Comments and Suggestions for Authors

The authors in the current study aimed to characterize the in vitro radiobiological response of prostate cancer (PCA) cell models to different radiation qualities (X-rays and 223Ra) in combination with inhibitors of key proteins of DDR system including an ATM inhibitor 95 (AZD0156), ATR inhibitor (AZD6738) and PARP inhibitor (AZD2281). They addressed knowledge gaps relative to metastatic castration-resistant Prostate cancer (mCRPC) which represents a challenge for urologists, oncologists, and radiotherapists.

The manuscript is easily readable and well-written. Moreover, the methodology results are robust. The controls of experiments were well assessed. The results add important advances to actual therapeutic options for mCRPC:

- First, they showed that the ATR inhibitor and the PARP inhibitor radiosensitized all prostate cancer cells, as previously stated (PMID= 35906379, 31197228);

- Second, the novelty of the observations relative to combinations of DDR inhibitors and 223Ra. Specifically, those combinations resulted in the greatest levels of apoptosis. It is an interesting result should be acknowledged at large. 

- Third, I should be careful to state "the effectiveness of current treatments for localized and metastatic castration-resistant prostate cancer". The localized PCa should not benefit from PARP inhibitors or such invasive strategies (https://uroweb.org/guidelines/prostate-cancer);

Figure and Tables read well. 

Comments on the Quality of English Language

Minor spelling checking should be assessed. for example:

- Introduction, Line 93: " to characterise"

- Trough the manuscript the term "localised". It's "localized".

Author Response

We thank the reviewer for taking the time to read the manuscript and provide comments. Please see below in bold response to comments.

  • Third, I should be careful to state "the effectiveness of current treatments for localized and metastatic castration-resistant prostate cancer". The localized PCa should not benefit from PARP inhibitors or such invasive strategies (https://uroweb.org/guidelines/prostate-cancer). This sentence has now been changed please see line 18 for changes.

    Comments on the Quality of English Language

    Minor spelling checking should be assessed. for example:

    - Introduction, Line 93: " to characterise". This sentence has been changed, please see line 93 for changes.

    - Trough the manuscript the term "localised". It's "localized". This has been corrected please see corrected term in line 18, line 44 and line 410.

Reviewer 3 Report

Comments and Suggestions for Authors

General comments:

The authors determined DDR inhibition as a promising strategy to increase the effectiveness of current treatments for localized and metastatic castration-resistant prostate cancer. The antiproliferation effects of DDR inhibitors/radium-223 are better than DDR inhibitors/X-rays in this study. They provide the survival, cell cycle, and apoptosis data for all comparison between DDR inhibitors/radium-223 and DDR inhibitors/X-rays.

Major comments:

1. Figure 4: Cell cycle analysis was determined by G1, S, and G2/M; however, the subG1 was not analyzed. Is there any data for subG1 in this study? SubG1 is an apoptosis-like indicator. If there are no subG1, then this description needs to be added.

2. Table 1: Linear quadratic parameters need to be mentioned in Materials and methods. What is the alpha and beta parameters?

3. Table 1 is only explained by one sentence in the result context as follows: “Radiobiological parameters for each of the survival curves in Figure 1 are presented in Table 1.”. It is not clear what the meaning of this information is. More descriptions are needed.

4. Figure 3: Many values for the meant 53BP1 foci per cell were provided. However, no image data was provided. At least, the representative image for one of these data with 53BP1 foci (positive and negative) needs to show.

5. Discussion: The apoptosis data only depends on the PARP cleavage western blotting. The apoptosis signaling, such as caspase 3, 8, and 9, needs to be discussed for future works.

Minor comments:

1. p values (Figure legends): p needs to be typed in italic fonts.

2. What is the unit for the numbers in Table 1?

3. Please provide the reference for combination indices calculation if this calculation has been reported.

4. 2.2: “Cells were 119 exposed to doses ranging from 0 to 0.5 Gy, in up to 50 ul of Xofigo added to 2 ml of cell 120 culture medium, with an exposure time of 24 h.” The culture dish size or well plate was not mentioned. Please add it.

Author Response

We thank the reviewer for taking the time to read the manuscript and provide comments. Please see below in bold response to comments.

Major comments:

  1. Figure 4: Cell cycle analysis was determined by G1, S, and G2/M; however, the subG1 was not analyzed. Is there any data for subG1 in this study? SubG1 is an apoptosis-like indicator. If there are no subG1, then this description needs to be added. In this study, no significant sub G1 at a 24 h timepoint was detected. This detail has been added to the manuscript at lines 338-339.
  2. Table 1: Linear quadratic parameters need to be mentioned in Materials and methods. What is the alpha and beta parameters? Additional information on the linear quadratic parameters has been added to the manuscript please see lines 152-153.
  3. Table 1 is only explained by one sentence in the result context as follows: “Radiobiological parameters for each of the survival curves in Figure 1 are presented in Table 1.”. It is not clear what the meaning of this information is. More descriptions are needed. Additional information has been added to the results section on table one, please see lines 207-209.
  4. Figure 3: Many values for the meant 53BP1 foci per cell were provided. However, no image data was provided. At least, the representative image for one of these data with 53BP1 foci (positive and negative) needs to show. Representative images for DU145 +/- X-rays and 223Ra +/- AZD2281 at 1 h timepoint have been added to the supplementary (supplementary figure 2) and are stated in the manuscript at lines 287-288.
  5. Discussion: The apoptosis data only depends on the PARP cleavage western blotting. The apoptosis signaling, such as caspase 3, 8, and 9, needs to be discussed for future works. This has been added to the conclusion section along with other future work. Please see lines 514-516.

Minor comments:

  1. p values (Figure legends): p needs to be typed in italic fonts. For figure legends for figure 2, 3 and 4 p is now in italics, please see highlighting in manuscript.
  2. What is the unit for the numbers in Table 1? The units for table one are presented in the table as α (Gy-1/-2) and β (Gy-1/-2).
  3. Please provide the reference for combination indices calculation if this calculation has been reported. This reference has now been added to the manuscript (ref 24) line 157.
  4. 2.2: “Cells were exposed to doses ranging from 0 to 0.5 Gy, in up to 50 ul of Xofigo added to 2 ml of cell 120 culture medium, with an exposure time of 24 h.” The culture dish size or well plate was not mentioned. Please add it. This information has now been added to the manuscript, please see line 120.

Reviewer 4 Report

Comments and Suggestions for Authors

Cell line research on the radiosensitising effect of PARP-inhibitors to Radium223 and Radiation.

Basic research with a strong proof of principle.

Hypothesis generating research.

Author Response

We thank the reviewer for taking the time to read the manuscript and providing positive feedback.

Round 2

Reviewer 3 Report

Comments and Suggestions for Authors

All reviewer's comments have been well responded.